# The Mental Fatigue Induced by Physical, Cognitive and Combined Effort in Amateur Soccer Players: A Comparative Study Using EEG

**DOI:** 10.3390/jfmk10040373

**Published:** 2025-09-27

**Authors:** Ana Rubio-Morales, Jesús Díaz-García, Marika Berchicci, Jesús Morenas-Martín, Vicente Luis del Campo, Tomás García-Calvo

**Affiliations:** 1Faculty of Sport Sciences, University of Extremadura, 10004 Cáceres, Spain; rubiomorales@unex.es (A.R.-M.); jesusmorenas@unex.es (J.M.-M.); viluca@unex.es (V.L.d.C.); tgarciac@unex.es (T.G.-C.); 2Department of Psychology, University ‘G. d’Annunzio’, 66100 Chieti, Italy; marika.berchicci@unich.it; 3Behavioural Imaging and Neural Dynamics (BIND) Center, University ‘G. d’Annunzio’, 66100 Chieti, Italy

**Keywords:** cognitive effort, mental fatigue, football, electroencephalography

## Abstract

**Objective:** Mental fatigue (MF) worsens soccer performance. Further knowledge is needed to understand MF’s effects on soccer players and its underlying mechanisms. Our aim was to analyze the subjective, objective, and neural MF-related outcomes induced by different type of tasks. **Methods:** A randomized crossover experimental design with repeated measures was used. Thirteen amateur soccer players (*M_age_* = 23 ± 5.43) completed three conditions: cognitive (30 min. Stroop.), physical (30 min. cycling), or combined (30 min. Stroop while cycling). Ratings of mental fatigue (measured via the Visual Analogue Scale), electroencephalographical signals (electroencephalography), and psychomotor performance (Brief-Psychomotor Vigilance Test) were measured pre- and post-condition. Soccer-related decision-making (TacticUP^®^ test) was assessed post-condition. **Results:** Linear Mixed Models analysis revealed increments in perceived mental fatigue in all conditions, especially cognitive (*p* = 0.004) and combined (*p* < 0.0001) conditions. Psychomotor performance worsened, especially for cognitive (*p* = 0.039) and combined (*p* = 0.009) conditions. The Individual Alpha Peak Frequency was lower after the cognitive task (*p* = 0.040) and compared with the physical task (*p* = 0.021). The Alpha midline power increased after the cognitive task in the central-frontal (*p* = 0.047) and central-posterior brain regions (*p* = 0.043). **Conclusions:** Cognitive and combined conditions were found to be more mentally demanding and fatiguing than single physical tasks. This was also reflected by an impaired reaction time. Based on the neural activity recorded, the performance impairments caused by mental fatigue were caused by reduced brain readiness (i.e., a lower Alpha Peak Frequency). However, non-significant changes were found in soccer-related decision-making. Coaches should consider the type of training tasks they recommend in light of their different effects on mental fatigue and performance.

## 1. Introduction

Soccer is not only physically demanding, but also cognitively demanding: specific game scenarios imply continuous vigilance, decision-making under time pressure, emotion management, and constant processing of internal and external information [1]. These high cognitive and emotional demands can lead to an exacerbated state of mental fatigue (MF) in soccer players [2]. MF is defined as a psychobiological state caused by prolonged or intense mental demands. MF can manifest throughout alterations that can be quantified subjectively (i.e., ratings of perceived effort) or objectively (i.e., reaction time and accuracy in psychomotor performance tests and decision-making tasks, and changes in heart rate variability and cortical electrical activity) [3,4]. There are individual factors that can influence the perception and consequences of MF (i.e., personality traits or accumulated experience) [3]. Therefore, it is necessary to move towards more holistic approaches when studying MF and its effects on athletic performance. 

The negative effects of mental fatigue on soccer performance have been widely demonstrated [5]. In soccer, MF has been found to negatively affect performance across various aspects of the game. Physically, it reduces the athlete’s ability to sustain effort by increasing the rating of perceived effort [6]. Regarding psychomotor performance, players experiencing MF tend to have slower and less accurate shots, commit more passing errors, maintain possession for shorter periods, and show weaker ball control and tackling success [7,8]. When it comes to decision-making, MF disrupts team coordination, leading to reduced lateral synchronization and smaller distances between defensive and midfield lines, which affects overall team spacing [6]. Additionally, it impairs both the speed and precision of decision-making during play [9]. Despite the strong evidence of the detrimental effects of MF on soccer performance, Thompson et al. (2019) suggested that further knowledge is needed to understand MF and mitigate its detrimental effects on soccer players’ health, well-being, and performance [2]. Considering that soccer players are involved in physical, cognitive, and combined tasks, this study aimed to analyze the effect of these different types of tasks on MF.

Previous research has examined the effect of different types of tasks (i.e., cognitive, physical, and mixed) on perceived MF in healthy, physically active subjects [10]. The results showed that all tasks increased the perception of MF, with higher levels in those with greater cognitive demands (i.e., cognitive and mixed tasks). Although this last study used a similar experimental protocol to the present study, our study aims to extends this methodology to a specific population (i.e., amateur soccer players). Furthermore, this study includes measurements of electroencephalography (EEG) to further research the possible neurophysiological markers of mental fatigue. In fact, most of the studies on MF in sport have used subjective instruments to measure it (i.e., the Visual Analogue Scale), as well as scales and tests aimed at measuring its subjective (i.e., rating of perceived effort) and behavioural effects (i.e., reaction time through the psychomotor vigilance test) [11]. Experts in sport psychology and movement science have highlighted the lack of information on the neurophysiological markers underlying MF [2].

Some studies have found increased alpha and theta spectral power after mentally fatiguing tasks, specifically in midline fronto-central and centro-parietal regions (i.e., FCz, Cz, CPz) [12]. Furthermore, Wascher et al. (2014) evidenced increases in alpha power for fronto-central (i.e., FCz) site, and in theta power for fronto-central and parieto-occipital (i.e., POz) sites after a long spatial stimulus–response-compatibility task [13]. Similar results were found by Boksem et al. (2005) after a 3 h visual attention task performed by healthy subjects [14]. Concretely, lower-alpha power was greatest on Pz, upper-alpha power was largest over occipital sites (i.e., O1, Oz, and O2), and theta power was greatest on frontal midline electrodes (i.e., Fz and Cz).

The present study aims to identify neurophysiological markers to further investigate the potential neurophysiological signs associated with MF. While subjective measures such as self-reported MF provide valuable insights, they have inherent limitations in capturing the complex and dynamic nature of MF. This limitation has driven the search for more objective neurophysiological markers. One such marker is Individual Alpha Peak Frequency (IAPF), an electrophysiological marker highly associated with cognitive processing and predisposition [15]. Higher IAPF values have been linked to better working memory [16], faster information processing, and shorter reaction times [17]. IAPF appears to reflect not only stable individual traits but also intra-individual state changes, such as increased cognitive demand or arousal [15]. However, to our knowledge, there are no studies analyzing the relationship between IAPF and MF.

While the use of neurophysiological techniques such as EEG is well-justified in the context of MF, the inclusion of behavioural variables like decision-making and psychomotor performance is equally critical. MF is not an isolated phenomenon, and it can also be manifested through other objective variables (e.g., a decline in response accuracy and response time while making decisions or performing psychomotor tasks) [5,18]. Decision-making is a core component of soccer performance, where players must process complex information and make rapid strategic choices. Similarly, psychomotor performance serves as a measure of the efficiency in maintaining selective attention and executing motor responses. Therefore, by including these variables, our study goes beyond mere neurophysiological assessment to provide a more holistic perspective on how MF affects the soccer player’s ability to perform.

Given that MF impairs executive functioning, which in turn affects decision-making and psychomotor performance, it is hypothesized that higher values of MF would be associated with a decrease in IAPF, since lower IAPF has been previously linked to reduced alertness and diminished cognitive efficiency [15].

Thus, the aim of the present study was to analyze the effects of cognitive (COG), physical (PHYS), and combined (COMB) (i.e., simultaneous cognitive and physical) tasks on MF in amateur soccer players. Specifically, we analyzed MF through its subjective (i.e., perceived MF) and objective outputs (i.e., psychomotor performance, and soccer-related decision-making), and neural variables (IAPF, alpha, and theta midline power spectra). It was hypothesized that all tasks would lead to increased levels of perceived MF, with the most cognitively demanding conditions (i.e., COG and COMB) eliciting the highest levels. Furthermore, greater impairments in psychomotor performance and decision-making were expected following the COG and COMB tasks, given their strong impact on executive functioning. Finally, the COG and COMB tasks were hypothesized to induce neurophysiological alterations, specifically a decrease in IAPF and an increase in midline alpha and theta power, reflecting the modulation of the brain activity after higher cognitive load.

## 2. Materials and Methods

### 2.1. Sample

Thirteen Spanish amateur soccer players (ten males and three females; *M_age_* = 23 5.43 years; *M_training/week_* = 4.08 3.82 h) voluntarily took part in this study. Exclusion criteria included the presence of injuries or mental health conditions that would prevent participants from completing physical and cognitive tasks. An a priori power analysis was conducted using the *simr* package in RStudio (version 2024.12.0+467), based on our planned linear mixed-effects model with two within-subject fixed factors (i.e., condition: COG, PHYS, COMB; and time: pre, post). Secondary endpoints included reaction time in the psychomotor performance test, neurophysiological markers (i.e., IAPF and midline alpha and theta powers), and response time/ratio scores in the soccer decision-making test. The assumed effect size for the protocol × time interaction was set to a medium effect (*f*^2^ = 0.25). The variance components, including participant-level random effects and pre–post correlations, were considered in the simulations. A total of 1000 simulations were conducted to improve the accuracy of the power analysis, with the primary endpoint defined as the change in perceived MF across the different task conditions. A significance level of α = 0.05 was applied to all tests. The analysis indicated that 65 participants would be required to achieve 80% power.

Considering the technical and logistical complexity of EEG data collection in sport settings, we obtained a final sample of 13 participants. With the final sample of 13 participants, and assuming the same model structure and effect size, a simulation-based post hoc analysis indicated that the actual statistical power was approximately 52.6% (95% CI: 49.5% to 55.7%). While this falls below the recommended threshold, the use of linear mixed-effects models with repeated measures helped to optimize statistical efficiency by accounting for within-subject correlations and leveraging multiple observations per participant, thereby enhancing sensitivity even in the context of a small sample [19].

### 2.2. Outcomes and Instruments

#### 2.2.1. EEG Recordings

EEG signals were digitally recorded using SennsLite software (v.1.2.1) (Bitbrain, Zaragoza, Spain). Sixteen semi-dry electrodes were placed in the Versatile EEG-16 channels cap model (Bitbrain, Zaragoza, Spain) in positions Fp1, Fp2, Fz, F3, F4, Cz, C3, C4, Pz, P3, P4, P7(T5), P8(T6), Oz, O1, and O2 of the 10–20 International Electrode Placement System, covering frontal, central, temporo-parietal, and occipital brain areas. Specifically, five frontal electrodes (i.e., Fp1, Fp2, F3, Fz, and F4), three central electrodes (i.e., C3, C4, and Cz), five temporo-parietal electrodes (P3, P4, Pz, P7(T5), and P8(T6)), and three occipital electrodes (O1, Oz, and O2) were used. EEG signals were sampled at 256 Hz and recorded with a common reference. The ground electrode was located at AFz and impedance values were below 10 KΩ. EEG was recorded pre- and post-protocol for 2 min with eyes closed, followed by 2 min with eyes open (i.e., looking straight ahead, at a fixed grey dot with a black background, delimited in the centre of a 23.8-inch, full HD 1920 × 1080 screen). This approach has been used in similar studies [15]. Although both eyes-open and eyes-closed resting-state EEG were recorded, only the eyes-closed condition was used for alpha power analysis. This decision was based on the well-documented dominance of alpha rhythms during eyes-closed conditions [20]. In contrast, theta power was analyzed under both eyes-open and eyes-closed conditions.

#### 2.2.2. Heart Rate

Polar Team Pro System (Polar Electro, Kempele, Finland) was used to measure heart rate in the PHYS and COMB protocols to ensure the maintenance of the intensity between 65% and 75% of the estimated maximum heart rate according to Tanaka et al. (2001) equation [21]. Based on this data (i.e., estimated maximum heart rate and age), the Polar Team Pro app (Polar Electro, Kempele, Finland) automatically generated personalized heart rate zones for each participant. During the sessions, participants could monitor their heart rate in real time on a 10.2-inch iPad (Apple Inc., Cupertino, CA, USA) placed in front of them through colour-coded feedback, allowing them to adjust their effort to remain within the target zone (65–75% of HRmax). Compliance was verified visually during the session and by later analysis of the percentage of time each participant spent within the designated zone. Sessions were considered valid when participants remained within the target intensity zone for at least 90% of the total session time.

#### 2.2.3. Incongruent Stroop Task

To induce MF, we employed a 30 min incongruent version of the Stroop colour-word task, which has been widely used as an effective method to elicit sustained cognitive effort and MF in both athletes and non-athlete populations [22]. The task consisted of the continuous presentation of colour words (i.e., “yellow,” “blue,” “green,” and “red”) printed in incongruent ink colours. Participants were instructed to name the ink colour of each word aloud, thereby requiring the inhibition of the automatic response to read the word itself.

This prolonged incongruent condition, characterized by sustained interference and response inhibition, is known to tax executive control and attention systems over time, thereby increasing MF. Stimuli were presented in a self-paced format using a paper-based version, and participants were continuously engaged throughout the 30 min task without breaks. Researchers monitored performance and provided real-time feedback: if a participant made an error, they were immediately asked to correct it before continuing. This procedure tried to ensure high cognitive engagement.

#### 2.2.4. Perceived Mental Fatigue

The Visual Analogue Scale (VAS-100 mm) was used to quantify participants’ perceived MF. This instrument consists of a 100 mm horizontal line anchored with “0 = Not at all mentally fatigued” on the left and “100 = Extremely mentally fatigued” on the right. Participants were asked to rate their level of MF immediately before and immediately after each protocol by marking a point along the line in response to the question “How mentally fatigued do you feel right now?” The distance in millimetres from the left end to the participant’s mark was recorded as their MF score. This is the most commonly subjective instrument used in MF research and has been reported as the most sensitive for detecting changes in acute perceived MF [8,11].

#### 2.2.5. Perceived Mental Load

After each protocol, participants reported their perception of mental load through the cognitive load item from the validated questionnaire to quantify the Mental Load in Team Sports (QMLST) [23]. Participants answered the question “How demanding would you quantify the cognitive effort of this task?” on a Likert scale from 0 (*not at all*) to 10 (*maximal*).

#### 2.2.6. Psychomotor Performance

The Brief-Psychomotor Vigilance Task (PVT-B) was employed to assess simple reaction time [24] using a mobile application (PVT Research Tool, Texas A&M University, College Station, TX, USA). Participants were instructed to tap the smartphone screen as soon as a red circle appeared at the centre for three minutes. The stimulus onset was randomized, occurring between 1 and 4 s. For each participant and trial, the average reaction time (in milliseconds) was computed from all valid responses within the 3 min test. This procedure was repeated before and after each experimental protocol. The modulation of reaction time in this task has been widely used as a behavioural indicator of acute MF, with increases in response latency typically interpreted as signs of cognitive impairment related to MF [25].

#### 2.2.7. Decision-Making in Soccer

The TacticUP^®^ platform (www.tacticup.com.br, accessed on 5 May 2025) was used to evaluate participants’ decision-making abilities. This video-based tool allows for an objective assessment of players’ tactical understanding in soccer. Its validity and reliability have been well established [26]. The test consists of random video clips showing both offensive and defensive situations in full 11 vs. 11 matches, recorded from a bird’s-eye view of the pitch. After each video, participants must select the most suitable option from four possible answers as fast as possible. Before starting the test, they receive instructions and complete three practice trials to become familiar with the format. The test generates scores for each of the 13 core tactical principles, along with separate scores for offensive and defensive phases, and an overall performance score. It also tracks the response time for each scenario. However, we calculated the response time/score ratio to run further data analysis, in order to obtain a more real context-related variable. The video test was performed through a computer with 23.8-inch, full HD (1920 × 1080) screen in standardized conditions, and it lasted around 20 min for each participant. TacticUP^®^ test was administrated in four occasions (i.e., after each protocol and before the start of the study, as baseline data for further analysis).

#### 2.2.8. EEG Analysis

EEG data were analyzed using BrainVision Analyzer software version 2.3.0.8301 (Brain Products GmBH, Gilching, Germany). The first and last five seconds of each recording were removed for the analysis. A band pass filter between 3 and 30 Hz was applied to the EEG data, together with a 50 Hz notch filter. Then, EEG data was semi-automatically inspected to identify artifacts (instrumental, ocular, and muscular) using the ICA ocular correction and the Raw Data Inspection features in BrainVision Analyzer software. Ocular artifacts were corrected using ICA (Infomax Extended algorithm; 512 steps; convergence bound = 1 × 10^–7^). Blink events were detected using the slope algorithm on the Fp1 channel referenced to F3. Components related to vertical ocular activity were identified based on the squared correlation with the vertical activity channel, and components accounting for 30% of the total squared correlation were removed from the data. Raw data inspection was then performed semi-automatically for all the 16 channels, and the following artifact rejection criteria were applied: (i) voltage step exceeding 50 µV/ms; (ii) amplitude difference greater than 100 µV in any 200 ms interval; and (iii) low activity below 0.5 µV within a 100 ms window. Segments were marked as bad if they exceeded any of these thresholds within a window from 200 ms before to 200 ms after the event, and they were excluded from further analysis. Continuous EEG data were then divided into 2 s segments with 50% overlap (i.e., new segments started every 1 s). EEG power spectra were then computed using the Fast Fourier Transform (FFT) with a frequency resolution of 0.5 Hz using the Welch method. According to the literature, the most MF-related frequency bands were selected: 4–8 Hz for the theta wave, and 8–12 Hz for the alpha wave (Ghojazadeh et al., 2024) [27].

To determine the IAPF, power spectra were computed using FFT across selected posterior electrodes in the eyes closed condition, as follows: three occipital (O1, O2, Oz) and five parietal (P7, P3, Pz, P4, P8) electrodes. These sites were chosen following [28], suggesting that the posterior cortex with the eyes closed offers the most reliable signal for estimating IAPF. The centre of gravity method within the 7.5–13.5 Hz band was used to calculate a weighted average frequency based on spectral power distribution [29]:$$\frac{\sum \left(a\left(f\right)\right)*f}{(\sum a(f))}$$

This technique captures the broader shape of the alpha activity and is particularly useful when multiple peaks are present, avoiding the bias of selecting only the most prominent one [29]. Note that this frequency range (i.e., 7.5–13.5 Hz) was used specifically for IAPF estimation, and not for general alpha power analyses (which used the conventional 8–12 Hz range). If a clear alpha peak was not observed within the selected range, no IAPF value was assigned for that case. However, all participants showed a clear alpha peak within the selected frequency range (7.5–13.5 Hz), allowing IAPF to be determined in all cases.

For alpha and theta midline powers, power spectrum data from FFT was used. We selected the individual average power (V^2^) spectra values for each frequency range (i.e., 8–12 Hz for alpha, and 4–8 Hz for theta) and from four midline electrodes (i.e., Fz, Cz, Pz, and Oz) of each test for the subsequent Linear Mixed Models (LMMs) analyses, based on previous research [30].

### 2.3. Procedures

A randomized crossover experimental design with repeated measures was carried out. Participants came to the lab at five different times (i.e., during the familiarization stage, a baseline session for the decision-making in soccer test, and the three protocols in a random order). The protocols performed were the following: (i) COG protocol (i.e., performing a paper version of the incongruent Stroop test for 30 min); (ii) PHYS protocol (i.e., riding on a cycloergometer maintaining the intensity between 65 and 75% of the individual estimated peak heart rate [21] for 30 min); and (iii) COMB protocol (i.e., cycling on a cycloergometer, with an intensity between 65 and 75% of their estimated peak heart rate, while simultaneously performing an incongruent Stroop test in paper form for 30 min) (see Figure 1). Continuous EEG signal, perceived MF, and psychomotor performance were measured before and after each protocol. Perceived mental load and decision-making in soccer were measured after each protocol. For the duration of the study, participants were encouraged to maintain their usual daily habits regarding caffeine intake, but they were encouraged not to ingest alcohol and to try to sleep well. Measurements were performed at the same time of day (between 4 and 9 pm) at least 48 h after a soccer training session and between sessions.

The participants were previously informed of the study procedures, and they all signed an informed consent form. The study was conducted in accordance with the ethical principles outlined in the Declaration of Helsinki (World Medical Association, 2013) and approved by the Ethics Committee for Biomedical Research of the University of Extremadura (protocol number: 113//2023). Data were treated in accordance with the ethical privacy codes of the American Psychological Association (2019).

### 2.4. Statistical Analysis

All data were analyzed using RStudio (version 2024.12.0+467) and the *dplyr*, *tidyr*, and *psych* packages for data manipulation and statistical calculations. Given the repeated measures design of the present study, where each participant was measured at multiple time points (pre- and post-protocols), a Linear Mixed Model (LMM) was chosen as the primary statistical approach. This method is a more flexible and robust alternative to a traditional two-way repeated measures ANOVA, particularly because it can handle data with missing values or unbalanced sample sizes across groups [19]. 

Prior to running the main analysis, a preliminary basic component analysis of variance was performed for each dependent variable to assess the contribution of the participant variable as a random factor. This step was crucial for justifying the use of a mixed model. The Wald Z statistic was used to test the null hypothesis that the population variance attributable to participants was zero. As the Intraclass Correlation Coefficient (ICC) was found to be greater than 10%, indicating significant variance between participants, the need to treat participants as a random effect in the LMM was corroborated [19].

Subsequently, LMMs were fitted using the *lme4* package to examine the effects of the protocols and times (pre- and post-protocols), which were considered fixed factors. The *participant ID* was included as a random intercept to account for the non-independence of the repeated measures within each subject. This approach models the individual baseline differences, allowing for a more accurate estimation of the fixed effects. These models were run on all dependent variables (i.e., EEG-related variables, perceived mental fatigue, perceived mental load, reaction time, and response time/score ratios in the TacticUP^®^ test).

To select the most appropriate model structure, a set of nested LMMs were compared. This process began with a null model that included only the random intercept and then progressed to more complex models that added the fixed effects of protocol, time, and finally, their interaction. Akaike’s Information Criterion (AIC) was used to compare these models, and in all cases, the model with the *protocol × time* interaction showed the lowest AIC value. This finding indicated that the interaction term significantly improved the model fit, and this final model was therefore used for all subsequent analyses. All statistical significance was set at a *p*-value of <0.05.

## 3. Results

Table 1 shows the descriptive statistics and main effects analyses of the performed LMM regarding the perceived (i.e., perceived MF and perceived cognitive load) and behavioural outcomes (i.e., reaction time in the PVT-B test) as dependent variables. A significant main effect of time (*F*(1, 72) = 115.33, *p* < 0.001) and protocol × time interaction (*F*(2, 72) = 7.05, *p* = 0.002) was observed for perceived MF. Post hoc comparisons (see Table 2) revealed significant increases in perceived MF after the COG (*p* < 0.001), PHYS (*p* < 0.05), and COMB (*p* < 0.001) protocols, with greater changes following protocols involving cognitive demands (i.e., COG and COM) compared to the PHYS protocol (*p* < 0.001). The model explained 66% of the variance in perceived mental fatigue (*R*^2^*m* = *R*^2^*c* = 0.66), indicating a large effect of the fixed predictors.

Regarding perceived cognitive load, the LMM showed a significant main effect of protocol (*F*(2, 24) = 52.17, *p* < 0.001). Post hoc comparisons (see Table 2) indicated that the cognitive load was significantly higher after the COG protocol compared to the PHYS protocol (*p* < 0.001), and also higher than after the COMB protocol (*p* < 0.05). The model explained 71% of the variance through fixed effects (*R*^2^*_m_* = 0.71) and 78% when including random effects (*R*^2^*c* = 0.78). Concerning the behavioural outcome (i.e., reaction time in the PVT-B test), the LMM revealed a significant main effect of protocol (*F*(2, 60) = 6.11, *p* = 0.004), but no significant effect of time (*F*(1, 60) = 2.35, *p* = 0.130) or protocol × time interaction (*F*(2, 60) = 3.02, *p* = 0.056). Post hoc analyses (see Table 2) showed that reaction time significantly increased after the COG protocol (*p* < 0.05), with higher values also observed in the COMB protocol compared to the PHYS one (*p* = 0.05). The model explained 13.8% of the variance through fixed effects (*R*^2^*_m_* = 0.14), and 48.2% when including random effects (*R*^2^*c* = 0.48), indicating that individual variability contributed substantially to the outcome.

Table 3 shows the descriptive statistics and main effects analyses of the run LMM for the IAPF. The model showed no significant main effect of protocol (*F*(2, 60) = 2.19, *p* = 0.121) or time (*F*(1, 60) = 0.72, *p* = 0.400), but a main effect of protocol × time interaction (*F*(2, 60) = 3.92, *p* = 0.031) was found. Post hoc comparisons revealed that IAPF significantly decreased after the COG protocol (*p* = 0.040) and when comparing it with the PHYS one (*p* = 0.021). The model explained only 0.6% of the variance through fixed effects (*R*^2^*_m_* = 0.006), while 95.3% was explained when including random effects (*R*^2^*c* = 0.95), indicating that most of the variability was attributable to inter-individual differences.

Regarding midline powers (see Table 3 and Table 4), alpha power at Fz revealed no significant main effect of protocol (*F*(2, 60) = 0.81, *p* = 0.449) or protocol × time interaction (*F*(2, 60) = 0.51, *p* = 0.603), but a significant main effect of time (*F*(1, 60) = 4.91, *p* = 0.030). Post hoc comparisons (see Table 5 and Table 6) showed significant increments after the COG protocol (*p* = 0.047). The model explained 10.8% of the variance through fixed effects (*R*^2^*_m_* = 0.108) and 59.7% when including random effects (*R*^2^*_c_* = 0.597). The LMM for alpha power at Pz showed no significant main effect of protocol (*F*(2, 60) = 0.74, *p* = 0.481), nor a protocol × time interaction (*F*(2, 60) = 0.43, *p* = 0.650). However, a significant main effect of time was observed (*F*(1, 60) = 4.32, *p* = 0.042). The variance explained by the fixed effects was 9.9% (*R*^2^*_m_* = 0.099), while the total variance explained by the full model was 70.1% (*R*^2^*_c_* = 0.701). Post hoc comparisons showed significant increments after the COG protocol (*p* = 0.043). No significant effects were observed for alpha midline powers at the remaining electrodes (i.e., Cz, and Oz) nor for theta midline powers across all electrodes (i.e., Fz, Cz, Pz, and Oz) and conditions (i.e., eyes closed and eyes open).

Finally, Table 7 displays the results of the LMM conducted on the normalized TacticUP^®^ response time/ratio scores (general, offensive, and defensive). No significant main effect of protocol (i.e., COG, PHYS, COMB, and baseline) was found for any of the variables: (i) For the general ratio (*F*(3, 36) = 0.685, *p* = 0.567, *R*^2^*_m_* = 0.021; *R*^2^*_c_* = 0.467); (ii) for the offensive ratio (*F*(3, 36) = 0.119, *p* = 0.948, *R*^2^*_m_* = 0.004, *R*^2^*_c_* = 0.491); and (iii) for the defensive ratio (*F*(3, 36) = 0.629, *p* = 0.601, *R*^2^*_m_* = 0.017, *R*^2^*c* = 0.531). Post hoc comparisons (see Table 8) did not reveal any significant differences.

## 4. Discussion

The aim of this study was to analyze the effects of COG, PHYS, and COMB (i.e., simultaneous cognitive and physical) tasks on MF (i.e., subjective, behavioural, and neural signs of MF) in amateur soccer players.

Regarding the subjective signs of MF, the present study showed significant increases in perceived MF after all the training tasks, with these changes being higher in the COG and COMB tasks than the PHYS task. In this sense, the first hypothesis has been verified. Our results are consistent with previous research [10], showing that cognitive tasks, particularly combined tasks (COG and COMB)**,** are the most mentally fatiguing. The Stroop test, which requires response inhibition, sustained attention, and working memory, further supports this finding, as it increases perceived MF. The higher cognitive load in the COG and COMB tasks compared to the PHYS task reflects the fact that MF arises from tasks with high or prolonged cognitive demands, regardless of whether these demands are physical or cognitive in nature [31]. The PHYS protocol also showed significantly higher perceived MF from pre- to post-measurements. The suggestions by Van Cutsem and Marcora (2021) about the influence of the maintenance of the pacing or self-confidence level during sports performance in MF could be a possible theoretical explanation, but there are only a few studies analyzing the effects of primarily physical tasks on MF [32]. Ref. [31] found that 15 min of cycling at a moderate intensity (i.e., level of 12–13 on Borg’s 6–20 RPE scale) after a cognitive demanding task showed a positive counter measurement in mentally exhausted individuals. Similar results were obtained by Oberste et al. (2021) after 30 min of cycling at a moderate intensity (i.e., 65–75% VO_2_ max) [33]. These authors attributed the effect of acute physical activity on the dopaminergic system (i.e., in favour of dopamine production) as a possible explanation for such improvements. However, other studies have reported reduced dopamine availability, diminished motivation, and impaired performance in mentally fatigued individuals [15], suggesting the need for further research.

Concerning the behavioural signs of MF, this study showed impairments in reaction time during psychomotor performance (i.e., PVT-B test) after the COG and COMPB tasks when compared with the PHYS task. Thus, these findings are in line with the initial hypothesis. Response time has been considered as a behavioural indicator of MF [18], and this is linked to the incorrect or delayed interpretation of visual stimuli, maladjusted movement responses, or delayed movement execution while performing mentally fatiguing tasks [34].

With reference to the neural variables, this is the first study to analyze EEG-related variables as indicators of MF depending on different training tasks in a sample of soccer players. IAPF showed significant decreases after the COG task and changes post–pre protocol were also significant compared with the PHYS task, which is partially in line with the initial hypothesis. IAPF has been evidenced as an important variable in cognitive processing [15], and its higher values in healthy subjects are correlated with better working memory capacity, speed of information processing, and better visual reaction times [16,17]. Conversely, a decline in IAPF has been suggested to be associated with MF, worse cognitive efficiency, and decreased vigilance [16]. Although there is not much research regarding the relationships between IAPF, MF and training tasks, the high mental demands present in the COG protocol seemed to have a negative effect on IAPF and, consequently, on the reaction time and sustained attention required to perform the PVT-B test. The higher cognitive load in the COG task likely resulted in a shift from top-down cognitive control [16], as the brain allocates more resources to maintain performance, leading to cognitive overload. This overload reduces the brain’s ability to sustain high-frequency alpha waves, which are essential for maintaining focus and filtering out irrelevant information. Overall, the IAPF findings are promising and suggest that IAPF could serve as a useful neurophysiological marker of MF.

Alpha midline power showed significant increments for Fz and Pz electrodes after the COG task, only partially reflecting what was suggested in the hypothesis. The systematic review and meta-analysis carried out by Tran et al. (2020) indicated the influence of mental fatigue in alpha and theta bands, incrementing the power values of frontal (i.e., Fz), central (i.e., Cz), and parieto-occipital sites (i.e., Pz, P3, P4, O1, and O2) [12]. Specifically, higher amplitudes of alpha waves in frontal brain regions have been associated with the “top-down” system (i.e., while demanding internal cognitive processing) and, consequently, with the onset of MF [35]. Thus, it is not surprising to find higher alpha midline powers after the most cognitively demanding protocol (i.e., the cognitive task).

Finally, no significant differences were found for response time/ratio scores in the soccer-related decision-making specific task, failing to confirm the initial hypothesis. Previous research has shown the negative effects of MF in decision-making performance in soccer. Smith et al. (2016) evidenced impairments in accuracy and speed of soccer-specific decision-making skill in a MF state, assessed through a video test in which players were asked to make the most appropriate decision regarding a central offensive midfielder [36]. Worse positioning and lateral synchronization between players were found by Coutinho et al. (2017) in mentally fatigued soccer player while performing small-sided games [6]. During soccer training matches (i.e., 90 min) after a high and prolonged cognitive task, Gantois et al. (2020) found decrements in passing decision-making performance [9]. The results of the present study showed impairments after all the protocols compared with the baselines (except for the offensive ratio after the COMB task), but with no significant differences. Raab and Laborde (2011) found that handball players who predominantly used intuitive decision-making strategies exhibited faster responses compared to those who adopted a more deliberative approach [37]. Although decision-making preferences were not measured in the present study, it could be possible that MF did not affect decision-making, since players could have been more intuitive than deliberative in their decisions. Future research should differentiate between the type of decision-making and monitoring the video test using EEG recordings (preferably ERP techniques) to better justify these results. This discrepancy could also be explained by factors such as the level of expertise, task sensitivity, and methodological differences. We suggest future studies explore more context-specific tasks and compare different levels of expertise to better capture these effects.

### 4.1. Limitations and Future Guidelines

A small sample of amateur soccer players participated in this study, and this was a limitation to achieve an adequate statistical power. Future research should include bigger samples of soccer players from different categories to compare the effects of different training tasks on MF depending on the expertise levels. Furthermore, our study employed non-specific tasks to induce MF, nor to measure some performance outcomes (i.e., response time through the PVT-B test). Nevertheless, it employed wide used and well-evidenced tests to induce MF and to measure its behavioural signs. Future studies should employ more ecological tasks and task related to the players’ daily life to induce MF (e.g., small-sided games, playing videogames, using smartphones) and to measure performance (e.g., validated soccer technical and tactical tests, penalty shoots, repeated sprint ability tests) [2,5]. EEG recordings represent an objective way to measure the mechanisms underlying MF and its effects on performance, but it is recommended to use other EEG recording techniques with higher temporal resolution (i.e., Event-Related Potentials). This technique allows us to analyze changes in brain activity during the performance of the tasks.

### 4.2. Practical Applications

Considering the results of the present study, there are some practical applications to be considered by coaches and staff. Firstly, it is important to consider the type of tasks included during training due to their different effects on MF and performance. Tasks with high cognitive demands, such as decision-making drills or complex tactical exercises, should be avoided near competition, as they may increase MF. Instead, cognitive tasks can be incorporated into mid-training days to help players adapt to MF without interfering with match preparation.

Another possibility is to include doses of Brain Endurance Training (BET) during field or gym sessions to enhance players’ endurance to MF. BET combines simultaneous physical and cognitive tasks, resulting in improvements in physical, cognitive, technical, and multitasking performance in soccer players [38]. This method could be beneficial for increasing resistance to MF, although more research is needed to confirm its long-term effects.

In addition, neurofeedback training has emerged as a promising method to improve athletic performance by targeting brain wave power [39]. By regulating brain wave activity, particularly in the alpha and theta bands, neurofeedback may improve cognitive functions, which could enhance resistance to MF during high-pressure moments in matches.

Furthermore, IAPF and alpha midline power could be used as tools for assessing MF and guiding interventions aimed at improving cognitive performance during training or competition. Real-time monitoring of these markers could help coaches adjust the intensity of cognitive tasks and prevent cognitive overload, ensuring that players are mentally prepared for competition.

Finally, when reaction time is crucial to performance, coaches should avoid using mentally fatiguing tasks prior to the match, as these can impair reaction times and decision-making speed during the game.

## 5. Conclusions

This study aimed to analyze the effects of cognitive, physical, and combined (i.e., cognitive and physical simultaneously) tasks on holistic MF. Perceived MF increased significantly after all the tasks, but the effects of cognitive and combined tasks on perceived MF were significantly greater compared with the physical task. As a behavioural indicator of MF, the reaction time in the PVT-B test was significantly higher after the cognitive task, and after the cognitive and combined tasks compared with the physical one. IAPF was significantly lower after the cognitive tasks and compared with the physical task. Alpha midline power in Fz and Pz electrodes was significant higher after the cognitive task. No significant effects on decision-making in soccer were found after any of the tasks. Coaches should consider the type of tasks they include during training due to the tasks’ different effects on mental fatigue and performance in soccer. Higher IAPF and alpha midline powers (i.e., fronto-central and parieto-central sites) could be considered as indicators of higher cognitive loads.

## Figures and Tables

**Figure 1 jfmk-10-00373-f001:**
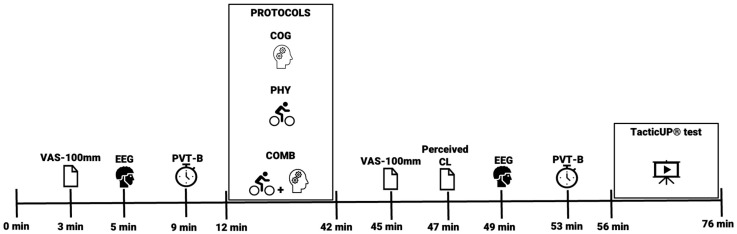
Study timeline and experimental procedure. Participants underwent three phases: (1) Familiarization (pre-study period), during which participants were introduced to the tasks and equipment; (2) baseline (pre-measurement phase), where initial measurements were recorded for the TacticUP^®^ test, and (3) experimental (task conditions phase), which included three task conditions (i.e., COG, PHYS, COMB) in a randomized counterbalanced order, with pre- and post-measures of perceived MF, reaction time in the PVT-B test, and IAPF and alpha and theta midline powers; and post-measures of the perceived cognitive load, and the TacticUP^®^ test for soccer decision-making. Note: VAS-100 mm—Visual Analogue Scale-100 mm for MF; EEG—electroencephalography recording (i.e., 2 min with eyes closed + 2 min with eyes open); PVT-B—3 min Brief Psychomotor Performance Test; COG—cognitive protocol (i.e., 30 min of Incongruent Stroop Test); PHY—physical protocol (i.e., 30 min cycloergometer at 65–76% of the estimated Peak Heart Rate); COMB—combined protocol (i.e., 30 min cycloergometer at 65–76% of the estimated peak heart rate while performing the Incongruent Stroop Test); CL—cognitive load, and TacticUP^®^ test—20 min soccer specific decision-making video test.

**Table 1 jfmk-10-00373-t001:** LMM descriptive statistics and main effects analyses for perceived (i.e., perceived MF and perceived cognitive load) and behavioural (i.e., reaction time) outcomes.

Variable	Protocol	Pre (Emmean ± SE)	Post (Emmean ± SE)	Main Effects	*p*-Value	$\eta_{p}^{2}$
Perceived MF (a.u.)	COG	41.50 ± 0.42	82.30 ± 0.42	Protocol: *F*(2, 72) = 12.06	<0.001 ***	0.290
	PHYS	33.80 ± 0.42	53.80 ± 0.42	Time: *F*(1, 72) = 155.33	<0.001 ***	0.710
	COMB	31.50 ± 0.42	83.10 ± 0.42	Protocol × Time Interaction: *F*(2, 72) = 7.05	0.002 **	0.230
Perceived Cognitive Load (a.u.)	COG	-	8.15 ± 0.32	Protocol: *F*(2, 24) = 52.17	<0.001 ***	0.840
	PHYS	-	4.08 ± 0.32			
	COMB	-	7.62 ± 0.32			
Reaction Time (ms)	COG	380 ± 8.80	403 ± 8.80	Protocol: *F*(2, 60) = 6.11	0.004 **	0.170
	PHYS	383 ± 8.80	374 ± 8.80	Time: *F*(1, 60) = 2.35	0.130	0.040
	COMB	396 ± 8.80	408 ± 8.80	Protocol × Time Interaction: *F*(2, 60) = 3.02	0.056	0.090

*Note.* emmean (estimated marginal mean); SE (standard error); a.u.: arbitrary units; COG (cognitive); PHYS (physical); COMB (combined). ** *p* < 0.01; *** *p* < 0.001.

**Table 2 jfmk-10-00373-t002:** Multiple comparisons (i.e., intra- and inter-condition) for perceived (i.e., perceived MF and perceived cognitive load) and behavioural (i.e., reaction time) outcomes.

	Perceived MF			Perceived Cognitive Load			Reaction Time		
Comparisons	*p*-Value	Cohen’s *d*	CI (95%)	*p*-Value	Cohen’s *d*	CI (95%)	*p*-Value	Cohen’s *d*	CI (95%)
**Inter-protocol (Pre)**									
COG vs. PHYS	0.714	0.183	[−0.07, 0.44]	-	-	-	0.998	−0.049	[−0.30, 0.20]
COG vs. COMB	0.443	0.239	[−0.02, 0.49]	-	-	-	0.512	−0.224	[−0.48, 0.03]
PHYS vs. COMB	0.998	−0.055	[−0.31, 0.20]	-	-	-	0.752	0.175	[−0.08, 0.43]
**Inter-protocol (Post)**									
COG vs. PHYS	<0.001 ***	0.679	[0.40, 0.96]	<0.001 ***	2.07	[1.38, 2.77]	0.039 *	0.392	[0.13, 0.65]
COG vs. COMB	1.00	−0.018	[−0.27, 0.23]	0.387	0.274	[−0.12, 0.67]	0.994	−0.070	[−0.32, 0.18]
PHYS vs. COMB	<0.001 ***	0.698	[0.42, 0.98]	<0.001 ***	1.80	[1.16, 2.43]	0.009 **	0.462	[0.20, 0.73]
**Intra-protocol**									
PRE vs. POST (COG)	<0.001 ***	−0.973	[−1.28, −0.67]	-	-	-	0.164	−0.313	[−0.57, −0.06]
PRE vs. POST (PHYS)	0.006 **	−0.477	[−0.74, −0.21]	-	-	-	0.917	0.129	[−0.12, 0.38]
PRE vs. POST (COMB)	<0.001 ***	−1.23	[−1.56, −0.90]	-	-	-	0.821	−0.158	[−0.41, 0.09]

*Note.* COG (cognitive); PHYS (physical); COMB (combined). * *p* < 0.05; ** *p* < 0.01; *** *p* < 0.001.

**Table 3 jfmk-10-00373-t003:** LMM descriptive statistics and main effects analyses for IAPF, and alpha and theta waves in eyes-closed condition on different sites.

Variable	Protocol	Pre (Emmean ± SE)	Post (Emmean ± SE)	Main Effects	*p*-Value	$\eta_{p}^{2}$
IAPF (Hz)	COG	9.95 ± 0.12	9.87 ± 0.13	Protocol: *F*(2, 60) = 2.19	0.121	0.070
	PHYS	9.94 ± 0.12	9.99 ± 0.13	Time: *F*(1, 60) = 0.72	0.400	0.010
	COMB	9.93 ± 0.12	9.91 ± 0.13	Protocol × Time Interaction: *F*(2, 60) = 3.92	0.031 *	0.080
α midline power-Fz (μV^2^)	COG	1.51 ± 0.43	1.86 ± 0.43	Protocol: *F*(2, 60) = 0.81	0.449	0.030
	PHYS	1.69 ± 0.43	1.89 ± 0.43	Time: *F*(1, 60) = 4.91	0.030 *	0.080
	COMB	1.78 ± 0.43	1.88 ± 0.43	Protocol × Time Interaction: *F*(2, 60) = 0.51	0.603	0.020
α midline power-Cz (μV^2^)	COG	1.51 ± 0.26	1.71 ± 0.26	Protocol: *F*(2, 60) = 0.27	0.766	0.009
	PHYS	1.61 ± 0.26	1.70 ± 0.26	Time: *F*(1, 60) = 2.53	0.117	0.040
	COMB	1.61 ± 0.26	1.77 ± 0.26	Protocol × Time Interaction: *F*(2, 60) = 0.13	0.879	0.004
α midline power-Pz (μV^2^)	COG	0.69 ± 0.15	0.84 ± 0.15	Protocol: *F*(2, 60) = 0.74	0.481	0.020
	PHYS	0.80 ± 0.15	0.86 ± 0.15	Time: *F*(1, 60) = 4.91	0.030 *	0.070
	COMB	0.74 ± 0.15	0.81 ± 0.15	Protocol × Time Interaction: *F*(2, 60) = 0.43	0.650	0.010
α midline power-Oz (μV^2^)	COG	2.37 ± 0.59	2.82 ± 0.59	Protocol: *F*(2, 60) = 4.76	0.062	0.140
	PHYS	2.17 ± 0.59	2.12 ± 0.59	Time: *F*(1, 60) = 0.28	0.598	0.005
	COMB	3.18 ± 0.59	3.23 ± 0.59	Protocol × Time Interaction: *F*(2, 60) = 0.29	0.745	0.010
θ midline power-Fz (μV^2^)	COG	0.74 ± 0.09	0.71 ± 0.09	Protocol: *F*(2, 60) = 2.08	0.134	0.060
	PHYS	0.95 ± 0.09	0.09 ± 0.09	Time: *F*(1, 60) = 0.11	0.744	0.002
	COMB	0.87 ± 0.09	0.86 ± 0.09	Protocol × Time Interaction: *F*(2, 60) = 0.003	0.997	0.009
θ midline power-Cz (μV^2^)	COG	0.62 ± 0.07	0.64 ± 0.07	Protocol: *F*(2, 60) = 1.15	0.322	0.040
	PHYS	0.74 ± 0.07	0.77 ± 0.07	Time: *F*(1, 60) = 0.03	0.851	0.006
	COMB	0.66 ± 0.07	0.71 ± 0.07	Protocol × Time Interaction: *F*(2, 60) = 0.14	0.868	0.005
θ midline power-Pz (μV^2^)	COG	0.56 ± 0.09	0.62 ± 0.09	Protocol: *F*(2, 60) = 0.793	0.457	0.030
	PHYS	0.61 ± 0.09	0.79 ± 0.09	Time: *F*(1, 60) = 0.088	0.768	0.001
	COMB	0.62 ± 0.09	0.72 ± 0.09	Protocol × Time Interaction: *F*(2, 60) = 0.185	0.832	0.006
θ midline power-Oz (μV^2^)	COG	0.44 ± 0.08	0.45 ± 0.08	Protocol: *F*(2, 60) = 1.82	0.170	0.060
	PHYS	0.56 ± 0.08	0.62 ± 0.08	Time: *F*(1, 60) = 0.32	0.574	0.005
	COMB	0.57 ± 0.08	0.65 ± 0.08	Protocol × Time Interaction: *F*(2, 60) = 0.18	0.838	0.006

*Note.* emmean (estimated marginal mean); SE (standard error); α (alpha); θ (theta); COG (cognitive); PHYS (physical); COMB (combined). * *p* < 0.05.

**Table 4 jfmk-10-00373-t004:** LMM descriptive statistics and main effects analyses for theta waves in eyes-open condition on different sites.

Variable	Protocol	Pre (Emmean ± SE)	Post (Emmean ± SE)	Main Effects	*p*-Value	$\eta_{p}^{2}$
θ midline power-Fz (μV^2^)	COG	0.74 ± 0.09	0.71 ± 0.09	Protocol: *F*(2, 60) = 5.88	0.467	0.160
	PHYS	0.95 ± 0.09	0.95 ± 0.09	Time: *F*(1, 60) = 0.08	0.782	0.001
	COMB	0.87 ± 0.09	0.86 ± 0.09	Protocol × Time Interaction: *F*(2, 60) = 0.02	0.023	0.007
θ midline power-Cz (μV^2^)	COG	0.62 ± 0.07	0.64 ± 0.07	Protocol: *F*(2, 60) = 3.15	0.502	0.090
	PHYS	0.74 ± 0.07	0.77 ± 0.07	Time: *F*(1, 60) = 0.53	0.469	0.009
	COMB	0.66 ± 0.07	0.71 ± 0.07	Protocol × Time Interaction: *F*(2, 60) = 0.04	0.959	0.001
θ midline power-Pz (μV^2^)	COG	0.56 ± 0.09	0.62 ± 0.09	Protocol: *F*(2, 60) = 1.12	0.333	0.040
	PHYS	0.61 ± 0.09	0.79 ± 0.09	Time: *F*(1, 60) = 2.99	0.089	0.050
	COMB	0.62 ± 0.09	0.72 ± 0.09	Protocol × Time Interaction: *F*(2, 60) = 0.31	0.732	0.010
θ midline power-Oz (μV^2^)	COG	0.44 ± 0.08	0.45 ± 0.08	Protocol: *F*(2, 60) = 3.44	0.386	0.100
	PHYS	0.56 ± 0.08	0.62 ± 0.08	Time: *F*(1, 60) = 0.85	0.360	0.010
	COMB	0.57 ± 0.08	0.65 ± 0.08	Protocol × Time Interaction: *F*(2, 60) = 0.13	0.878	0.004

*Note.* emmean (estimated marginal mean); SE (standard error); θ (theta); COG (cognitive); PHYS (physical); COMB (combined).

**Table 5 jfmk-10-00373-t005:** Multiple comparisons (i.e., intra- and inter-condition) for IAPF, and alpha and theta waves in eyes-closed condition on different sites.

	**IAPF**			**α midline power-Fz**			**α midline power-Cz**			**α midline power-Pz**			**α midline power-Oz**		
**Comparisons**	** *p* ** **-value**	**Cohen’s *d***	**CI (95%)**	** *p* ** **-value**	**Cohen’s *d***	**CI (95%)**	** *p* ** **-value**	**Cohen’s *d***	**CI (95%)**	** *p* ** **-value**	**Cohen’s *d***	**CI (95%)**	** *p* ** **-value**	**Cohen’s *d***	**CI (95%)**
**Inter-protocol (Pre)**															
COG vs. PHYS	0.999	0.027	[−0.22, 0.28]	0.905	−0.133	[−0.38, 0.12]	0.986	−0.084	[−0.33, 0.16]	0.709	−0.184	[−0.44, 0.07]	0.998	0.052	[−0.20, 0.30]
COG vs. COMB	0.995	0.068	[−0.18, 0.32]	0.610	−0.205	[−0.46, 0.05]	0.989	−0.081	[−0.33, 0.17]	0.988	−0.082	[−0.33, 0.16]	0.562	−0.214	[−0.47, 0.04]
PHYS vs. COMB	0.996	−0.041	[−0.29, 0.21]	0.993	0.071	[−0.18, 0.32]	1.000	−0.003	[−0.25, 0.25]	0.968	−0.102	[−0.35, 0.15]	0.321	0.266	[0.01, 0.52]
**Inter-protocol (Post)**															
COG vs. PHYS	0.021 *	−0.376	[−0.63, −0.12]	1.000	−0.025	[−0.28, 0.22]	1.000	0.006	[−0.24, 0.25]	1.000	−0.023	[−0.27, 0.22]	0.707	0.185	[−0.07, 0.44]
COG vs. COMB	0.961	−0.107	[−0.36, 0.14]	1.000	−0.021	[−0.27, 0.23]	0.998	−0.052	[−0.30, 0.20]	0.999	0.046	[−0.20, 0.30]	0.956	−0.110	[−0.36, 0.14]
PHYS vs. COMB	0.311	−0.269	[−0.52,−0.01]	1.000	−0.004	[−0.25, 0.25]	0.994	0.059	[−0.19, 0.31]	0.994	−0.069	[−0.32, 0.18]	0.217	0.295	[0.04, 0.55]
**Intra-protocol**															
PRE vs. POST (COG)	0.040 *	0.256	[0.07, 0.51]	0.047 *	−0.262	[−0.52,−0.01]	0.823	−0.158	[−0.41, 0.09]	0.043 *	−0.251	[−0.51, 0.03]	0.939	−0.119	[−0.37, 0.13]
PRE vs. POST (PHYS)	0.862	−0.147	[−0.40, 0.10]	0.837	−0.154	[−0.41, 0.10]	0.995	−0.067	[−0.32, 0.18]	0.385	−0.255	[−0.53, 0.05]	1.000	0.014	[−0.24, 0.26]
PRE vs. POST (COMB)	0.989	0.081	[−0.17, 0.33]	0.989	−0.079	[−0.33, 0.17]	0.914	−0.130	[−0.38, 0.12]	0.930	−0.123	[−0.37, 0.13]	1.000	−0.014	[−0.26, 0.24]
	**θ midline power-Fz**			**θ midline power-Cz**			**θ midline power-Pz**			**θ midline power-Oz**					
**Comparisons**	** *p* ** **-value**	**Cohen’s *d***	**CI (95%)**	** *p* ** **-value**	**Cohen’s *d***	**CI (95%)**	** *p* ** **-value**	**Cohen’s *d***	**CI (95%)**	** *p* ** **-value**	**Cohen’s *d***	**CI (95%)**			
**Inter-protocol (Pre)**															
COG vs. PHY	0.670	−0.193	[−0.45, 0.06]	0.829	−0.156	[−0.41, 0.09]	0.978	−0.093	[−0.34, 0.16]	0.995	−0.068	[−0.32, 0.18]			
COG vs. COMB	0.981	−0.090	[−0.34, 0.16]	1.00	0.012	[−0.24, 0.26]	0.999	−0.030	[−0.28, 0.22]	0.773	−0.170	[−0.42, 0.08]			
PHY vs. COMB	0.968	−0.102	[−0.35, 0.15]	0.788	−0.086	[−0.34, 0.16]	0.996	−0.063	[−0.31, 0.19]	0.968	0.102	[−0.15, 0.35]			
**Inter-protocol (Post)**															
COG vs. PHY	0.733	−0.179	[−0.43, 0.07]	0.994	−0.070	[−0.32, 0.18]	0.988	−0.081	[−0.33, 0.17]	0.999	0.035	[−0.21, 0.29]			
COG vs. COMB	0.985	−0.086	[−0.34, 0.16]	1.00	0.015	[−0.23, 0.27]	0.993	0.072	[−0.18, 0.32]	0.864	−0.147	[−0.40, 0.10]			
PHY vs. COMB	0.979	−0.093	[−0.34, 0.16]	0.985	−0.085	[−0.34, 0.17]	0.841	−0.153	[−0.40, 0.09]	0.720	0.182	[−0.07, 0.43]			
**Intra-protocol**															
PRE vs. POST (COG)	1.000	0.019	[−0.23, 0.27]	1.000	−0.016	[−0.27, 0.23]	1.000	−0.016	[−0.27, 0.23]	0.986	−0.085	[−0.34, 0.17]			
PRE vs. POST (PHY)	0.999	0.032	[−0.22, 0.28]	0.994	0.070	[−0.18, 0.32]	1.000	−0.004	[−0.25, 0.25]	1.000	0.019	[−0.23, 0.27]			
PRE vs. POST (COMB)	1.000	0.023	[−0.23, 0.27]	1.000	−0.011	[−0.26, 0.24]	0.985	0.086	[−0.16, 0.34]	0.997	−0.061	[−0.31, 0.19]			

*Note.* COG (cognitive); PHYS (physical); COMB (combined). * *p* < 0.05.

**Table 6 jfmk-10-00373-t006:** Multiple comparisons (i.e., intra- and inter-condition) for theta waves in eyes-open condition on different sites.

	θ Midline Power-Fz			θ Midline Power-Cz			θ Midline Power-Pz			θ Midline Power-Oz		
Comparisons	*p*-Value	Cohen’s *d*	CI (95%)	*p*-Value	Cohen’s *d*	CI (95%)	*p*-Value	Cohen’s *d*	CI (95%)	*p*-Value	Cohen’s *d*	CI (95%)
**Inter-protocol (Pre)**												
COG vs. PHY	0.231	−0.290	[−0.55, −0.03]	0.523	−0.222	[−0.48, 0.03]	0.997	−0.060	[−0.31, 0.19]	0.785	−0.167	[−0.42, 0.09]
COG vs. COMB	0.710	−0.184	[−0.44, 0.07]	0.991	0.077	[−0.33, 0.17]	0.992	−0.075	[−0.33, 0.18]	0.772	−0.171	[−0.42, 0.08]
PHY vs. COMB	0.962	−0.106	[−0.36, 0.14]	0.870	−0.145	[−0.40, 0.11]	1.000	0.015	[−0.24, 0.26]	1.000	0.003	[−0.25, 0.25]
**Inter-protocol (Post)**												
COG vs. PHY	0.126	−0.329	[−0.59,−0.07]	0.461	−0.235	[−0.49, 0.02]	0.624	−0.202	[−0.45, 0.05]	0.509	−0.225	[−0.48, 0.03]
COG vs. COMB	0.636	−0.199	[−0.45, 0.05]	0.919	−0.128	[−0.38, 0.12]	0.932	−0.122	[−0.37, 0.13]	0.335	−0.263	[−0.52,−0.01]
PHY vs. COMB	0.914	−0.130	[−0.38, 0.12]	0.961	−0.107	[−0.36, 0.14]	0.989	−0.079	[−0.33, 0.17]	0.999	0.038	[−0.21, 0.29]
**Intra-protocol**												
PRE vs. POST (COG)	0.999	0.039	[−0.21, 0.30]	0.999	−0.033	[−0.28, 0.22]	0.996	−0.066	[−0.32, 0.18]	1.000	−0.019	[−0.27, 0.23]
PRE vs. POST (PHY)	1.000	−0.001	[−0.25, 0.25]	0.999	−0.046	[−0.30, 0.20]	0.596	−0.208	[−0.46, 0.04]	0.991	−0.076	[−0.33, 0.17]
PRE vs. POST (COMB)	1.000	0.023	[−0.23, 0.27]	0.987	−0.084	[−0.33, 0.17]	0.950	−0.113	[−0.36, 0.14]	0.955	−0.111	[−0.36, 0.14]

*Note.* COG (cognitive); PHYS (physical); COMB (combined).

**Table 7 jfmk-10-00373-t007:** LMM descriptive statistics and main effects analyses for soccer-related decision-making outcomes.

Variable	Protocol	Descriptive Statistics (Emmean ± SE)	Main Effects	*p*-Value	$\eta_{p}^{2}$
Response Time/Ratio Score (Total Actions)	COG	0.431 ± 0.103	Protocol: *F*(3, 36) = 0.685	0.567	0.050
	PHYS	0.547 ± 0.103			
	COMB	0.460 ± 0.103			
	BS	0.402 ± 0.103			
Response Time/Ratio Score (Offensive Actions)	COG	0.450 ± 0.096	Protocol: *F*(3, 36) = 0.119	0.948	0.010
	PHYS	0.470 ± 0.096			
	COMB	0.418 ± 0.096			
	BS	0.428 ± 0.096			
Response Time/Ratio Score (Defensive Actions)	COG	0.464 ± 0.086	Protocol: *F*(3, 36) = 0.629	0.601	0.050
	PHYS	0.551 ± 0.086			
	COMB	0.485 ± 0.086			
	BS	0.441 ± 0.086			

*Note.* emmean (estimated marginal mean); SE (standard error); COG (cognitive); PHYS (physical); COMB (combined); BS (baseline).

**Table 8 jfmk-10-00373-t008:** Multiple comparisons (i.e., intra- and inter-condition) for soccer-related decision-making outcomes.

	Response Time/Ratio Score (Total Actions)			Response Time/Ratio Score (Offensive Actions)			Response Time/Ratio Score (Defensive Actions)		
Comparisons	*p*-Value	Cohen’s *d*	CI (95%)	*p*-Value	Cohen’s *d*	CI (95%)	*p*-Value	Cohen’s *d*	CI (95%)
**Inter-protocol (Pre)**									
COG vs. PHYS	0.703	−0.180	[−0.50, 0.14]	0.997	−0.034	[−0.36, 0.29]	0.734	−0.171	[−0.49, 0.15]
COG vs. COMB	0.993	−0.044	[−0.37, 0.28]	0.987	0.056	[−0.27, 0.38]	0.994	−0.042	[−0.36, 0.28]
PHYS vs. COMB	0.848	−0.135	[−0.46, 0.19]	0.948	−0.091	[−0.41, 0.23]	0.864	−0.130	[−0.45, 0.19]
COG vs. BS	0.993	0.046	[−0.28, 0.37]	0.995	0.039	[−0.28, 0.36]	0.993	0.046	[−0.28, 0.37]
COMB vs. BS	0.948	0.090	[−0.23, 0.41]	0.999	−0.017	[−0.34, 0.30]	0.952	0.087	[−0.23, 0.41]
PHYS vs. BS	0.534	0.226	[−0.10, 0.55]	0.971	0.074	[−0.25, 0.40]	0.567	0.217	[−0.11, 0.54]

*Note.* COG (cognitive); PHYS (physical); COMB (combined); BS (baseline).

## Data Availability

The data presented in this study are available on request from the corresponding author.

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
