# Peer review of "The Mental Fatigue Induced by Physical, Cognitive and Combined Effort in Amateur Soccer Players: A Comparative Study Using EEG"

_jfmk, 2025, doi:10.3390/jfmk10040373_

Round 1

Reviewer 1 Report

Comments and Suggestions for Authors

The manuscript titled " The Mental Fatigue Induced by Physical, Cognitive and 2 Combined Effort in Amateur Soccer Players: A Comparative 3 Study Using EEG" has been of interest in recent decades in sports, and the use of techniques such as EEG provides great value for its scientific contribution. The integration of subjective and neurophysiological measures is a strength that enriches the interpretation of results and provides novel evidence to the applied field of sports performance.

Some suggestions for the authors to consider:

Title: accurate and interesting. Includes the population, variables, induction conditions, and assessment instrument.

Abstract: suggests improvement.

- This section indicates that this is an experimental study; however, the procedure states that it is a quasi-experimental study (page 3, line 131).

- An abbreviation of MF, for mental fatigue, is used in the abstract without indicating its origin.

- Soccer performance is included in the keywords. The sample is amateur soccer. Inconsistency. Are soccer performance?? Are amateur soccer?? Are the same?

- Include in this section and in parentheses the tests performed to evaluate each variable evaluated. For example: mental fatigue (assessment instrument), psychomotor performance (assessment instrument), etc.

Introduction.

- This is a solid section, describing the variables and presenting the most relevant contributions made to date in this field. However, the main suggestion for the authors is to further compare previous research that has used the same experimental protocol (e.g., Rubio-Morales et al., 2022) with the present manuscript, highlighting the similarities, differences, and the specific contribution of this study. Furthermore, the use of techniques such as EEG in this context is widely justified; however, the use of variables included in this study, such as decision-making, psychomotor performance, etc., is lightly addressed.

Methodology.

- Sample. The wording related to the procedure, such as the Declaration of Helsinki, the Bioethics Committee, etc., is included. It is suggested that the procedure be included in the corresponding section dedicated to this purpose.

- Procedures. It is important that authors consistently adhere to methodological terminology: the abstract describes the study as experimental, while this section presents it as quasi-experimental. This section is proposed to be changed and appear after the outcomes and instruments sections.

- The remaining sections are detailed, and no modifications are suggested.

Results.

- It is recommended to review the presentation of Table 1. It is confusing and makes it difficult to correctly view the variable, protocol, and inter-protocol columns together.

- It is also suggested to review Tables 3 and 4. Align the estimated marginal mean of the Physical protocol. Similarly, the data appears to be incorrectly transferred (.0955) from the POST column and the variable response time/ratio score (total actions).

- In this section, it is noted that, in some cases, the values ​​have two decimal places, and in others, three. It is suggested to unify the criteria and maintain the same format throughout the text.

Discussion.

- The discussion constitutes one of the manuscript's most accomplished sections, as it coherently interweaves the results obtained with the previously established theoretical framework. The authors manage to contextualize their findings in relation to the existing scientific literature and, at the same time, highlight the practical applications of their results in the field of training and preparation of amateur soccer players. In this sense, this section adds solidity to the work and increases its potential impact on the scientific and professional community.

Author Response

Please find attached the document with the responses to your valuable comments.

Reviewer 2 Report

Comments and Suggestions for Authors

Overall, this manuscript addresses a timely and relevant topic, exploring the effects of mental fatigue on amateur soccer players using both subjective and neurophysiological measures. The integration of EEG markers such as IAPF and alpha power represents a valuable and innovative approach, and the study has the potential to contribute meaningfully to the field of sports science. The paper is well structured, and the authors are transparent in acknowledging key limitations, which strengthens the credibility of the work.

At the same time, there are several areas where the manuscript could be further improved to enhance its clarity, rigor, and impact. These include refining the introduction to ensure smoother transitions, providing a more transparent explanation of the sample size calculation and statistical analysis, reorganizing the presentation of results for greater interpretability, and deepening the critical analysis in the discussion. Minor but important issues, such as figure captions and reference formatting, also need to be addressed to align with the standards of the journal.

These suggestions are intended to strengthen an already interesting and relevant study. With clearer presentation, greater consistency, and a more in-depth interpretation of the findings, this work has the potential to make a valuable contribution to the literature on mental fatigue and athletic performance.

  • Lines 42–46: the use of ellipses to complete enumerations reduces the scientific rigor of the writing. Expressions such as “electrical activity from cortex, …” or “experience, personality, …” suggest incompleteness and convey an informal tone that is not appropriate for an academic manuscript. In scientific writing, precision and clarity are expected, and ellipses often give the impression of unfinished ideas or a lack of specificity. Overall, avoiding ellipses in this context would improve the academic quality of the manuscript, ensuring that each statement appears deliberate and complete rather than informal or unfinished.

  • Introduction: In the section introducing the Individual Alpha Peak Frequency (IAPF), the paragraph appears highly technical and somewhat abrupt in relation to the preceding content. While the inclusion of IAPF is relevant and scientifically justified, the transition into this concept could be smoother. At present, it jumps directly into technical details without first clarifying its practical significance for the study population. The introduction would benefit from a brief transitional explanation that highlights why IAPF is a particularly promising marker in the context of mental fatigue among athletes. For example, the authors could first connect the general limitations of subjective measures with the need for neurophysiological markers, and then position IAPF as a potential solution due to its established association with cognitive efficiency, processing speed, and reaction time.

  • Sample size: The sample size and power analysis, while conceptually appropriate (simulation-based power for a linear mixed model), is under-specified and therefore difficult to appraise. The manuscript does not report the assumed effect size for the protocol × time interaction, the variance components (including the participant-level random effects and pre–post correlations), or the covariance structure used in the simulations. Moreover, only 100 simulations per sample size were run; for Monte Carlo accuracy, a much larger number (e.g., ≥1000) is generally recommended. The analysis also appears to target α=0.05 without a clearly defined primary endpoint despite multiple outcomes, which risks overstating the effective power. 

  • Design: In the abstract, the study is described as “experimental,” while in the methods section it is referred to as “quasi-experimental.” This inconsistency in terminology is confusing and should be addressed. Based on the procedures reported, the design more closely aligns with a randomized crossover experimental design with repeated measures, rather than a quasi-experimental study. Although the sample is small and there is no parallel control group, the randomization of the order of conditions and the within-subject comparisons are clear experimental features. It is important that the terminology remains consistent throughout the manuscript. Referring to the study as both experimental and quasi-experimental undermines clarity and may mislead readers. I recommend that the authors standardize their description and consistently use the term “randomized crossover experimental design with repeated measures,” which accurately reflects the methodological approach employed.

  • Figures: Regarding Figure 1, the current caption is too brief and does not provide enough information for readers to fully understand the study timeline on its own. I suggest that the authors expand the description to include a clear explanation of each phase (familiarization, baseline, and experimental protocols), the timing of pre- and post-measures, and the instruments applied. This would ensure that the figure is self-contained, as expected in MDPI journals. As a reference, the authors may consult the style of figure descriptions used in other MDPI publications, such as the paper available at: https://www.mdpi.com/2411-5142/10/3/333. In that article, figures are accompanied by captions that are detailed enough to guide the reader without requiring them to return to the main text. Adopting a similar approach here would greatly improve the clarity and readability of Figure 1.

  • Statistical analysis section (lines 282–295): would benefit from clearer explanation and greater transparency. While the choice of linear mixed models (LMM) is methodologically sound for repeated measures data, the description is presented in a very technical way and may be difficult for readers without a strong statistical background to follow. For instance, the reference to a “basic component analysis of variance” and the use of the Wald Z statistic are not explained in sufficient detail. Similarly, it is not clear how the authors selected and compared model structures (e.g., covariance assumptions, handling of multiple comparisons, use of Akaike’s Information Criterion). A more explicit link between the LMM approach and its equivalence to a two-way repeated measures ANOVA would make the rationale more accessible. In addition, several abbreviations are introduced without being written out in full the first time they appear (e.g., ICC, AIC, LMM, R²m, R²c). For clarity and consistency, these terms should be defined upon first use—for example, Intraclass Correlation Coefficient (ICC). 

  • Results: To improve the organization and clarity of the results, I suggest restructuring the tables as follows. Tables 1 and 3 should present the descriptive statistics for each condition and time point, together with the results of the main analyses, specifically reporting the F values, p values, and partial eta squared as a measure of effect size. This would provide readers with a clear overview of the primary findings in a format similar to that of a two-way repeated measures ANOVA summary. 

    Tables 2 and 4 could then be dedicated exclusively to the multiple comparisons, both intra- and inter-condition. In these tables, it would be advisable to include not only the p values but also an effect size index, such as Cohen’s d, for each pairwise comparison. This structure would separate the main statistical outcomes from the detailed post-hoc results, making the findings easier to interpret and more aligned with best practices in scientific reporting.

  • Discussion: this section addresses the main hypotheses and successfully links the findings with previous research, particularly regarding the stronger impact of cognitive and combined tasks on perceived mental fatigue and psychomotor performance. The inclusion of EEG outcomes such as IAPF and alpha power is also valuable, and the authors are transparent in acknowledging key limitations such as the small sample size and the use of non-specific tasks.

    However, the discussion could be strengthened in several ways. At present, many of the arguments primarily confirm previous findings without offering a deeper critical analysis. For example, while prior studies reported impairments in soccer-specific decision-making under mental fatigue, the current study did not find significant effects, yet the explanation offered is brief and could be expanded with alternative interpretations (e.g., level of expertise, task sensitivity, or methodological differences). Similarly, the interpretation of the IAPF findings is promising but remains underdeveloped; further exploration of the neurophysiological mechanisms involved and their implications for applied contexts would enhance the contribution.

    In addition, the writing style is sometimes repetitive, particularly when emphasizing that cognitive tasks were more demanding. The authors could streamline these sections to make the discussion more concise and impactful. Finally, while the practical applications are outlined later, the discussion itself could integrate more applied perspectives to highlight the relevance of the findings for training and performance. Emphasizing the novelty of employing neurophysiological markers such as IAPF and alpha power as indicators of mental fatigue in athletes would further underline the originality and value of this work.

  • Cites Format: The in-text citations are not in the correct MDPI format. According to the journal’s guidelines, references should be cited using square brackets in the order of appearance (e.g., [1,2]) rather than superscript numbers. In the current version, references are formatted in superscript, which does not comply with MDPI requirements. The authors should revise all in-text citations accordingly to ensure consistency with the journal’s reference style.

Author Response

(The authors gave the same response as above.)

Round 2

Reviewer 2 Report

Comments and Suggestions for Authors

I would like to thank the authors for their careful and thorough revision of the manuscript. Overall, the changes introduced have substantially improved the clarity, methodological rigor, and scientific quality of the work.